

# Do popular apps have issues regarding energy efficiency?

Cagri Sahin

Department of Computer Engineering, Gazi University, Ankara, Turkiye

## ABSTRACT

Mobile apps have become essential components of our daily lives, seamlessly integrating into routines to fulfill communication, productivity, entertainment, and commerce needs, with their diverse range categorized within app stores for easy user navigation and selection. User reviews and ratings play a crucial role in app selection, significantly influencing user decisions through the interplay between feedback and quantified satisfaction. The emphasis on energy efficiency in apps, driven by the limited battery lifespan of mobile devices, impacts app ratings by potentially prompting users to assign low scores, thereby influencing the choices of others. In this study, the presence of energy consumption issues within widely-used popular apps that have high app ratings and user interaction has been investigated through the analysis of user reviews. It is anticipated that popular apps, with high ratings, are less problematic than other apps. User reviews were collected from 32 apps across 16 diverse categories and subsequently filtered based on specific keywords. From the resulting pool of 14,064 user reviews, 8,007 reviews were manually identified as specifically addressing the app's energy consumption. The results of the study demonstrate that all 32 popular apps under consideration exhibit issues related to energy consumption. While the frequency of energy-related issues may vary, it is evident that users are concerned about app energy consumption, as evidenced by the reception of complaint reviews regarding their energy usage. App energy efficiency is important to users, including popular apps with diverse features, necessitating developers to address expectations and optimize for energy efficiency. Improving the energy efficiency of apps has the potential to enhance user satisfaction and, consequently, contribute to the overall success of the app.

## INTRODUCTION

In the era of digital advancement, the pervasive use of mobile devices, particularly smartphones and tablets, has led to an exponential rise in the significance of mobile apps. These apps have seamlessly integrated into our daily routines, serving as indispensable tools for communication, productivity, entertainment, and commerce. The dynamic landscape of mobile applications, ranging from free utilities to paid premium services, is organized within app stores' diverse categories, simplifying user navigation and selection. For example, the Google Play Store holds the position of being the largest app store in terms of app availability, offering around 3.55 million apps that fall into 49 categories for Android users (*Statista, 2022*).

Corresponding author
Cagri Sahin, cagrisahin@gazi.edu.tr

When selecting apps from various categories, the significance of user reviews and ratings in shaping user decisions cannot be overstated (*Mahmood, 2020*; *Sällberg, Wang & Numminen, 2022*). This interplay between feedback and quantified satisfaction shapes user choices, consequently exerting a significant influence on users. For example, 85% of users avoid downloading apps with a two-star rating, while only 50% are interested in downloading apps with a three-star rating (*Alchemer, 2022*).

One of the factors that affects app ratings is the growing emphasis on energy efficiency within apps due to the limited battery lifespan of mobile devices. Energy efficiency issues in apps pertain to problems resulting in excessive energy consumption, inefficient resource utilization, and adverse effects on device battery life (*Wilke et al., 2013*). Such energy inefficiencies can prompt rapid battery drainage, directly undermining the usability of mobile devices. As a result, users can uninstall the app, choose another alternative app, and leave a low app score, thereby affecting other users' decisions.

In this study, I have investigated whether widely-used and well-known popular apps have issues related to energy consumption based on user reviews, as popular apps often have extensive user interactions and are expected to be less problematic than other apps. To answer the research question, 32 popular apps considered from 16 different app categories have been chosen among popular apps based on the criteria of having ratings of at least 4.0, more than 3 million reviews, and more than 100 million downloads from the Google Play Store. With these selection criteria, it is ensured that the chosen apps have high user satisfaction and substantial user interaction. User reviews were meticulously collected and subsequently filtered based on specific keywords, including "battery", "energy", and "power". To achieve more precise results, a total of 14,064 user reviews containing the specified keywords from the collected app reviews were manually analyzed, and 8,007 reviews specifically addressing the app's energy consumption were categorized as either negative or positive user reviews.

The study's findings reveal that users are indeed concerned about app energy consumption, often mentioning it in their reviews. Furthermore, the study highlights that even popular apps have received complaint reviews concerning their energy consumption, resulting in a negative impact on user experience. According to user reviews, all 32 considered popular apps exhibit energy consumption related issues. Additionally, users also express satisfaction with app energy consumption when they are content, often indicated by giving high rating scores. Therefore, improving app energy efficiency has the potential to enhance user satisfaction and, consequently, contribute to the overall success of the app. However, it's observed that, with the exception of specific apps, a fluctuating trend with variations characterizes app energy consumption issues. Interestingly, among the top five apps with the highest number of negative reviews, four exhibit a decreasing trend in energy consumption issues. Finally, energy consumption-related issues are less likely to be only linked to app categories and even with app functionalities since it is not consistently accurate with expectations.

The remainder of this article is organized as follows: the next section introduces background information. Then, the methodology of the study, including details of the study design and data retrieval process, is presented. This is followed by the presentation of

data processing, data analysis, results, and a discussion of findings regarding the research questions. Subsequently, the potential threats to validity and related work are examined in detail. Finally, a summary of the overarching insights gained from the study along with the direction of future research is provided.

## BACKGROUND

Mobile devices such smart phones and tablets use has significantly increased in the digital age, making mobile apps a necessary component of our daily life. Users increasingly rely on mobile apps to accomplish a wide range of tasks, including communication, productivity, entertainment, and shopping. Depending on the app's business model and the developer's plan, mobile apps can be either free or paid. To help organize and categorize the enormous number of apps available, app stores offer app categories, enabling users to navigate and find apps that meet their needs and interests.

When selecting an app within a specific app category, app reviews and ratings can play significant role in the user's decision-making process (*Mahmood, 2020*; *Sällberg, Wang & Numminen, 2022*). While app reviews provide valuable feedback from users who have firsthand experience with the app, app ratings provide a quantitative evaluation of the user satisfaction. With app reviews, users express their experiences, opinions, and suggestions regarding the app in addition to reporting both positive and negative aspects, issues and bugs. Developers can improve their app and enhance the user experience by analyzing the app reviews, identifying common themes and patterns, and using this feedback to make data-driven decisions for updates, bug fixes, feature enhancements, and overall app optimization (*Khalid, Nagappan & Hassan, 2016*; *Noei, Zhang & Zou, 2021*). On the other hand, users assign a rating based on their overall experience the app, usually on a scale ranging from 1 to 5 stars. While a five-star rating represents the highest level of satisfaction, a one-star rating represents the lowest level of satisfaction. For example, users give a five-star rating when they have had an excellent experience with the app and consider the app to be high quality. A three-star rating often suggests that users have an average or neutral experience with the app.

According to a recent report, app star ratings are extremely influential in the assessment of a app, as 90% of consumers considering them to be crucial (*Alchemer, 2022*). Users check app ratings and read reviews before downloading apps in the proportion of 79%, before updating apps in the proportion of 53%, and before making in-app purchases in the proportion of 55%. Moreover, the decision of a user to download or not to download an app is significantly influenced by app ratings and reviews. For example, 85% of users do not consider downloading apps with a two-star rating, and just 50 percent consider to download apps with three-star rating.

Given the significant impact of app ratings on user decision-making when it comes to downloading apps, it is crucial to thoroughly analyze the reasons behind low ratings. Low app ratings can be result of various factors such as bugs and technical issues, poor user experience, lack of features or functionality, performance problems, inadequate customer support, excessive ads, and a lack of updates or improvements. Another reason for low app ratings can be the energy efficiency issues in apps since the energy consumption of apps

has become a major concern for users due to the limited battery lifetimes of the mobile devices (*Wilke et al., 2013*). Energy efficiency issues in apps refer to problems that cause excessive energy consumption and inefficient use of device resources, and negatively impact the device's battery life. Consequently, energy inefficiencies in apps can cause rapid battery drain, which directly affects the usability of mobile devices. Some well known energy efficiency issues in mobile apps include continuous background activities, excessive network usage, inefficient CPU utilization, non-optimized screen usage and app display settings, unnecessary utilization of resources and hardware components, misuses of energy greedy APIs such as wakelock API, and inadequate power management (*Li et al., 2023*; *Sahin, Pollock & Clause, 2019*; *Sun et al., 2023*). For example, if an app acquires, but fails to release, energy-consuming resources such as GPS, these resources continue to stay in a high-power state and consume excessive amounts of energy.

## METHODOLOGY

This section describes the details of the study design, including considered keywords, application store, and applications in addition to the data retrieval process.

### Keywords

There are several common techniques used in app review analysis such as sentiment analysis, text mining, and keyword analysis. In this study, I mainly focused on keyword based analysis that could provide valuable insights into what users are talking about. To identify and analyze application energy consumption related app reviews by leveraging keywords, it is required to determine the most relevant problem-specific keywords that indicate energy issues or problems that users may be encountering with the app. For this purpose, previously identified energy-related keywords were obtained from existing studies (*Wilke et al., 2013*; *Phong et al., 2015*).

Table 1 provides a list of the keywords that are anticipated to be used in energy-related reviews. The first column, Keywords, refers the energy-related keywords which are battery, power, and energy. However, searching these keywords alone in the reviews can lead to false positives since the review may contain off-topic information. For example, the following review includes the energy keyword although it is not related to the energy consumption of the specific application being discussed: *"I only got this App to keep in touch with family in Australia. I've spent the past hour making requests for a verification code to set up my account. I've got nothing, either by text or email. I can't help but feel that these people just don't care. -Utter waste of time and energy."*. To address this issue, adjunct words, such as those in the second column of Table 1, can be used in combination with the keywords. Stars in the adjunct words indicate a wildcard character that can represent one or more characters in the words (*e.g.*, drain battery, draining battery, drains battery). Although the use of adjunct words can provide additional context and help reduce the risk of false positives, there is still a chance of such events occurring. As shown in the previous example, the adjunct word "wast*" is used in conjunction with the keyword "energy". Therefore, instead of solely relying on keywords with adjunct words to extract app energy-related reviews, a preferred approach involves combining keyword-based analysis

**Table 1 Keywords.**

| Keywords | Adjunct words |
|---|---|
| Battery | consum*, drain*, usage, kill*, hungry |
| Power | eat*, save, affect*, draw*, discharg* |
| Energy | heavy on, to* much, low*, wast* |

Note:
Asterisks (*) in the adjunct words indicate a wildcard character that can represent one or more characters in the words (*e.g.* drain battery, draining battery, drains battery).

with the manual analysis of reviews that contain keywords. This approach ensures the extraction of more accurate data by eliminating irrelevant information.

## Application store

There are different app stores available for users to download and install applications, such as Google Play Store, Apple App Store, and Amazon App Store. As the third quarter of 2022, Android users are able to download approximately 3.55 million apps from the Google Play Store, which makes it the biggest app store in terms of the number of available apps (*Statista, 2022*). As the same period of time, the Apple Store is the second-largest app store with approximately 1.6 million available apps for IOS and The Amazon Appstore is the third-largest app store with around 480,000 available apps for Android users.

Besides the number of available apps, Android-based mobile devices have a larger market share compared to IOS-based mobile devices. In the fourth quarter of 2022, Android have a global market share of over 71.8%, while IOS have a market share of around 27.6% (*Statista, 2023*). This means that the majority of smartphones sold worldwide run on the Android operating system.

In this study, Google Play Store was selected to investigate energy-related app reviews by leveraging keywords since it is one of the most popular app stores globally and widely used by Android smartphone users. It also allows users to leave detailed reviews and provide ratings for the apps they have downloaded. Moreover, it supports a variety of features that can help in the analysis of app reviews, such as filtering by language and sorting by most recent reviews. Overall, Google Play Store provides a large and diverse pool of reviews that are suitable for extracting information that is relevant to the specific aspects.

Although the Google Play Store does provide information on the number of app reviews and downloads for an app worldwide, the app ratings can vary by region due to differences in user behavior, preferences, and needs, as well as regional competition. For example, while Zoom app had a rating of 4.2 in the US region when the data was gathered, it had a rating of less than 4.0 in the Turkey region. Note that, all the data gathered for an app in this study is from the Google Play Store in the US.

## Applications

This study has been conducted by leveraging popular apps, which refer to widely used and well-known mobile applications. The reason for choosing popular apps as the subject is based on several factors. First, popular apps can be considered reliable because they are often supported by a team of dedicated developers from large tech companies. These

 

**Table 2 Considered applications (M: million, B: billion, +: more than)**

| Application name | Category | Ratings | Reviews | Downloads |
|---|---|---|---|---|
| Adobe Acrobat Reader | Productivity | 4.6 | 5.34M | 500M+ |
| AliExpress | Shopping | 4.5 | 13.7M | 500M+ |
| Amazon Prime Video | Entertainment | 4.1 | 3.77M | 500M+ |
| Avast antivirus & security | Tool | 4.7 | 7.09M | 100M+ |
| Canva | Art & Design | 4.8 | 11M | 100M+ |
| CapCut | Video Players & Editors | 4.4 | 5.63M | 500M+ |
| Duolingo | Education | 4.5 | 14.3M | 100M+ |
| eBay | Shopping | 4.7 | 4.48M | 100M+ |
| Firefox | Communication | 4.6 | 4.83M | 100M+ |
| Google Chrome | Communication | 4.1 | 41.9M | 10B+ |
| Google Maps | Travel and local | 4.1 | 16.7M | 10B+ |
| Google Photos | Photography | 4.5 | 47.1M | 5B+ |
| Instagram | Social | 4.0 | 143M | 1B+ |
| Messenger | Communication | 4.1 | 86.2M | 5B+ |
| Microsoft Teams | Business | 4.7 | 6.35M | 100M+ |
| Microsoft Word | Productivity | 4.5 | 7.06M | 1B+ |
| Netflix | Entertainment | 4.4 | 13.8M | 1B+ |
| Opera Browser | Communication | 4.3 | 4.3M | 100M+ |
| Piscart AI Photo Editor | Photography | 4.1 | 11.7M | 1B+ |
| Pinterest | Lifestyle | 4.6 | 9.63M | 500M+ |
| Snapchat | Communication | 4.2 | 31.9M | 1B+ |
| Spotify | Music & Audio | 4.4 | 27.9M | 1B+ |
| Telegram | Communication | 4.3 | 12M | 1B+ |
| TikTok | Social | 4.4 | 54.7M | 1B+ |
| Twitch | Entertainment | 4.4 | 5.12M | 100M+ |
| Twitter | Social | 4.0 | 20.8M | 1B+ |
| Uber | Map & Navigation | 4.7 | 11.3M | 500M+ |
| Wattpad | Book and Reference | 4.1 | 4.68M | 100M+ |
| Waze Navigation & Live Traffic | Map & Navigation | 4.4 | 8.62M | 100M+ |
| WhatsApp Messenger | Communication | 4.3 | 173M | 5B+ |
| YouTube | Video Players & Editors | 4.2 | 147M | 10B+ |
| Zoom | Business | 4.2 | 4.01M | 500M+ |

companies may invest a lot of funding for developing robust apps. Second, popular apps are frequently updated to fix bugs and address any issues that may arise, as well as to maintain their popularity and user satisfaction. Although these apps may have been well-tested before public releases, it is still possible bugs or other issues arise after the release. Third, popular apps can receive a significant amount of app reviews from users due to their widespread usage and visibility. By monitoring the feedback, developers can quickly address the users' concerns and complaints. Overall, it can be expected that popular apps might be less problematic than other apps.

**Table 3 Examples of app categories.**

| Category | Examples |
| --- | --- |
| Art & Design | Sketchbooks, painter tools, art and design tools, coloring books |
| Book & Reference | Book readers, reference books, textbooks, dictionaries, thesaurus, wikis |
| Business | Document editor/reader, package tracking, remote desktop, email management, job search |
| Communication | Messaging, chat/IM, dialers, address books, browsers, call management |
| Education | Exam preparations, study-aids, vocabulary, educational games, language learning |
| Entertainment | Streaming video, movies, TV, interactive entertainment |
| Lifestyle | Style guides, wedding and party planning, how-to guides |
| Map & Navigation | Navigation tools, GPS, mapping, transit tools, public transportation |
| Music & Audio | Music services, radios, music players |
| Photography | Cameras, photo editing tools, photo management, and sharing |
| Productivity | Notepad, to-do list, keyboard, printing, calendar, backup, calculator, conversion |
| Shopping | Online shopping, auctions, coupons, price comparison, grocery lists, product reviews |
| Social | Social networking, check-in |
| Tool | Tools for Android devices |
| Travel & local | Trip booking tools, ride-sharing, taxis, city guides, local business information, trip management tools, tour booking |
| Video Players & Editors | Video players, video editors, media storage |

Table 2 lists the specific applications that are being considered in this study. The first two columns, Application Name and Category, list the name of each application and the category to which it belongs, respectively. The third column, Ratings, shows the overall rating given by users for each application. The final columns, Reviews and Downloads, shows the number of times each app has been downloaded and reviewed at the time right before data collection for each app, respectively. While the review count indicates the total number of user reviews that have been submitted for the app, the download count represents the total number of times the app has been downloaded.

These apps have been chosen among popular apps based on the criteria of having ratings of at least 4.0, more than 3 million reviews, and more than 100 million downloads from the Google Play Store. These selection criteria can indicate that an app is well-received by users and is likely to be of high quality. For example, the TikTok app has a 4.4 rating, 54,7 million reviews, and over one billion downloads. A large number of reviews can also provide a diverse range of opinions and perspectives from different users.

Note that, the 32 selected apps come from 16 different categories on the Google Play Store, including Communication, Social, and more. Table 3 gives the examples of each category that are provided by the *Google Play Store (2023)*. Depending on the category an app belongs to, users may prioritize different aspects of the app. For example, apps in Map & Navigation category typically use GPS, which can consume a significant amount of energy. Therefore, it might be expected for users to leave reviews related to the energy consumption or battery drain of apps in this category. In contrast, users reviews for apps in education category may be less likely to include comments related to energy consumption or battery drain since they are not typically known to be heavy consumers of energy, as

they generally do not require intensive processing or constant use of device sensors like GPS. Analyzing reviews from apps in different categories can be useful in identifying user priorities and common themes related to energy consumption and efficiency.

### Data retrieval

To obtain the necessary app review data, a tool called App Review Collector (ARC) have been developed for Android App Store. The ARC tool provides various features including the ability to retrieve the review data written in a specific language, collect a specific number of most recent reviews, and save data in different file format such as json and csv.

For each considered app that is being analyzed, a variety of data points such as the date and the content of the review, the user's rating, as well as any replies made by the app developer along with their respective time and content were collected *via* the ARC tool and the language for review data is set to English. The computer used to collect review data was configured with Intel i5-2400 CPU 3.10 GHz, 8 GB of DDR3 memory, a 500 GB 7200 RPM SATA disk drive, and runs on Windows 10. During the data retrieval process, the upper limit of reviews collected per app was set to 500,000 due to the limitations in computer resources, the memory-intensive nature of the process, and the variability in app review data sizes. Additionally, app review data was considered within a range of 5 years (between January 2018 and January–February 2023) to ensure that the collected reviews are relevant and reflect the current state of the app since the age of the app can vary greatly.

Table 4 provides information about the collected review data for each app in January–February 2023. The first column, Application Name, lists the name of each application. The second column, Collected Reviews, shows the number of reviews limited to either the most recent 500,000 reviews or the last 5 years, whichever comes first. Note that, the collected data includes reviews exclusively in the English language. Third column, Beginning of Reviews, indicates the earliest user review date for the collected data. For example, the Zoom app has a total of 500,000 user reviews, which are collected starting from November 2022. On the other hand, the Twitch app has 244,240 user reviews, which are collected starting from January 2018. Also, there is an exceptional case for the CapCut app that sets it apart from other applications, once it is carefully examined. The CapCut app has less than 500,000 user reviews and its earliest user review date is in April 2020. The reason behind this is the first release date of the app is in April 2020. In total, 13,818,950 user reviews for 32 apps have been collected.

## DATA ANALYSIS AND DISCUSSION

I refined my overall question of whether popular applications, which are widely used and well known by a significant number of users, have energy related issues into the following specific research questions:

- *RQ1—Energy issue.* Do user reviews of popular apps indicate app energy consumption-related issues?
- *RQ2—Correlation.* Is there a correlation between the given app ratings and the reviews related to app energy consumption?

**Table 4 Collected user review data.**

| Application name | Collected reviews | Beginning of reviews | Keyword reviews | Energy reviews |
|---|---|---|---|---|
| Adobe Acrobat Reader | 309,203 | Jan-18 | 254 | 173 |
| AliExpress | 459,722 | Jan-18 | 524 | 268 |
| Amazon Prime Video | 500,000 | Mar-19 | 415 | 191 |
| Avast Antivirus & Security | 152,278 | Jan-18 | 955 | 651 |
| Canva | 500,000 | Oct-18 | 204 | 22 |
| CapCut | 375,309 | Apr-20 | 120 | 25 |
| Duolingo | 500,000 | Aug-20 | 312 | 46 |
| eBay | 466,786 | Jan-18 | 382 | 110 |
| Firefox | 189,415 | Jan-18 | 667 | 445 |
| Google Chrome | 500,000 | Apr-22 | 443 | 276 |
| Google Maps | 500,000 | Mar-21 | 563 | 302 |
| Google Photos | 500,000 | Apr-22 | 217 | 110 |
| Instagram | 500,000 | Oct-22 | 203 | 85 |
| Messenger | 500,000 | Jan-22 | 358 | 202 |
| Microsoft Teams | 500,000 | Jul-22 | 1,284 | 991 |
| Microsoft Word | 485,006 | Jan-18 | 240 | 41 |
| Netflix | 500,000 | May-22 | 520 | 165 |
| Opera Browser | 220,784 | Jan-18 | 240 | 172 |
| Piscart AI Photo Editor | 500,000 | May-19 | 172 | 54 |
| Pinterest | 500,000 | May-19 | 254 | 106 |
| Snapchat | 500,000 | Jan-22 | 366 | 242 |
| Spotify | 500,000 | Nov-21 | 700 | 392 |
| Telegram | 500,000 | Feb-21 | 367 | 193 |
| TikTok | 500,000 | Jul-22 | 390 | 237 |
| Twitch | 244,240 | Jan-18 | 343 | 251 |
| Twitter | 500,000 | Jan-21 | 522 | 155 |
| Uber | 500,000 | Jan-20 | 348 | 105 |
| Wattpad | 238,958 | Jan-18 | 143 | 73 |
| Waze Navigation & Live Traffic | 177,249 | Jan-18 | 897 | 689 |
| WhatsApp Messenger | 500,000 | Sep-22 | 214 | 100 |
| YouTube | 500,000 | Oct-22 | 220 | 74 |
| Zoom | 500,000 | Nov-22 | 1,217 | 1,061 |

- *RQ3—Category and functionality.* How do user reviews, especially those with high negative ratios regarding energy consumption, align with expectations based on app categories and functionality?
- *RQ4—Trend.* Is there a changing trend in app energy consumption issues according to the collected reviews?

To gather the data necessary to answer my research questions, the keywords "battery", "energy", and "power" were scanned in the collected app reviews for each app. The fourth column in the Table 4, Keyword Reviews, represents the number of user reviews that

include the considered keywords within the review text. For example, a total of 1,217 user reviews were found to contain the specified keywords for the Zoom app. However, there is a possibility that some of these keywords might be used in other contexts not directly related to energy consumption concerns. To ensure the accuracy of the data, user reviews that are not related to app energy consumption were discarded from the dataset such as *"I have ordered a small size of battery. It's being take 40–60 days to deliver to Sri Lanka. Disappointed."*. To achieve this, each review was manually analyzed. Although manual analysis is a time-consuming process, it plays a crucial role to ensure the accuracy and precision of the dataset. A keyword, even when used with adjunct words, might be employed to convey a different meaning depending on the context of the user review. For example, the following two user reviews include the terminology "power hungry", which is used in different meanings. In *"Deleting. I'll be back when this app isn't owned by a racist, power hungry prick."* user review, the phrase "power hungry" implies that the person in question seeks power and influence over others in an aggressive or self-serving manner, which is beyond the intended scope. Another user review states, *"This app is great but its too power hungry. Updated versions seems to be a bit better on power consumption."*. At this time, the phrase "power hungry" implies excessive energy usage or demands of the app. In addition to that, sometimes a user review might provide a false positive impression as seen in the following example: *"It's crashes my realme x7 max, android phone. It renders it in a non restartable state. I have to wait for the phone to drain its battery and the power it on. This problem occurs only in Netflix app, i have not faced this problem in any other app."* While this review may initially appear to be related to app energy consumption due to the mention of battery drain, it is more likely focused on an app crashing issue. The user expresses frustration about the app causing their phone to crash and become non-restartable. The mention of battery drain is a consequence of the app crashing, rather than a direct comment on its energy consumption. In yet another instance of a false positive example, *"Got rid of a virus that killed my battery 10/10."*, the user's satisfaction with the app is unrelated to its energy consumption. Instead, the primary focus of the user's review is on the app's claimed effectiveness in removing a virus that was causing battery issues on their device. Moreover, user reviews such as *"Battery type of work."* which do not make any sense, were removed from the dataset. The fifth column in Table 4, Energy Reviews, represents the number of user reviews that are specifically related to energy consumption concerns. For example, out of the 1,217 user reviews, 1,061 of them specifically address energy consumption concerns for the Zoom app. Overall, a total of 14,064 user reviews containing the considered keywords were thoroughly investigated, and out of these 8,007 reviews were found to be directly related to the app's energy consumption.

The remainder of this section discusses the results of my study in terms of the research questions, based on the dataset created by manually analyzing the user app reviews.

### RQ1: energy issue

App reviews for mobile applications, especially concerning energy consumption, are diverse in nature and may mainly focus on issues or problems encountered by users. In addition to that, positive reviews may also encountered in the feedback in case of user

satisfaction with the app's energy efficiency. Therefore, to answer the first research question, app reviews in the dataset need to be categorized as either negative or positive.

My initial attempt was to rely on the review scores, which are on a scale of 1 to 5. However, this option has a major drawback. The review score and the review content may not be correlated for the investigated aspect. The app might have energy consumption-related issues, but it can still receive a high score, as seen in the review example, *"Works well drains battery."* since users' evaluations, considerations, and preferences can vary. The user complains about the energy consumption of the app, but the given score was 4. The other option was to perform sentiment analysis, which required providing explicit lists of positive and negative words to determine the sentiment of the text. However, even with such positive and negative word lists, its effectiveness is limited when applied to domain-specific language, such as app energy consumption reviews. For example, the reviews *"Good, battery usage is very high."* and *"Best app ever... other than battery consuming."* can be categorized as positive while *"Seems to have fixed the severe battery drain issue."* can be categorized as negative using sentiment analysis. An alternative option was to utilize artificial intelligence APIs, which offer pre-trained models for sentiment analysis. However, a significant drawback of this approach is that most of these APIs come with a cost, making them less accessible for free usage.

As a result, I have decided to manually tag the energy consumption-related app reviews as positive and negative. This approach will not only provide more precise sentiment analysis results but also enable the creation of a labeled dataset for future research. Having a labeled dataset will be beneficial for training and evaluating sentiment analysis models specific to app energy consumption reviews and improving their accuracy. Note that while negative categorization includes user reviews that contain complaints, point out missing features, or make comparisons with other apps in a negative manner, positive categorization consists of reviews expressing appreciation and making comparisons with other apps in a positive manner in terms of app energy consumption. App comparisons in the reviews are often done with competitor apps, such as those that offer similar features or functionalities, for example, Firefox, Google Chrome, and Opera Browser.

Figure 1 shows the percentage distribution of negative and positive app reviews in terms of energy consumption for each app. While the x-axis represents the app names, the y-axis represents the percentage scale in the figure. The red (values above the y-axis 0) and green (values below the y-axis 0) colors are used for negative and positive reviews, respectively. Based on the Fig. 1, it is observed that all the popular apps analyzed are facing energy consumption-related issues from the user's perspective. The high percentage of negative reviews indicates that users have expressed concerns or dissatisfaction, specifically regarding energy consumption. For 21 out of 32 apps, the negative review percentage is more than 94%, and Uber, Snapchat, and TikTok apps have the highest negative review percentage. Out of the remaining 11 apps, four apps have a negative review percentage between 89% and 94%, five apps have a negative review percentage between 82% and 87%. For the Opera and Spotify apps, there are notable exceptions regarding negative review percentages. Despite having more negative reviews than positive ones, they stand out from other apps due to their higher positive review percentages. This suggests that while they

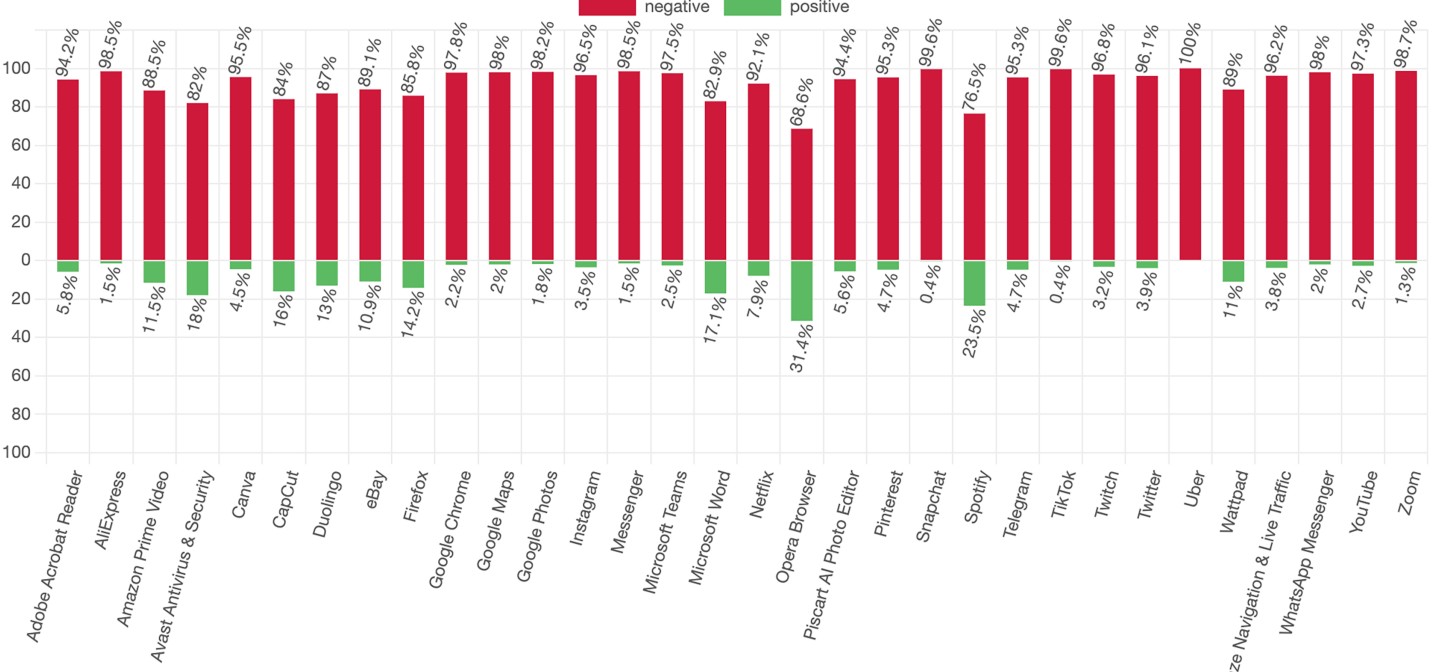

**Figure 1 Percentage distribution of negative and positive reviews for each app.** The red (values above the x-axis 0) and green (values below the x-axis 0) colors are used for negative and positive reviews, respectively.

**Table 5 Percentage distribution of app ratings for negative and positive reviews.**

| Rating | Negative reviews (%) | Positive reviews (%) |
|---|---|---|
| 5 | 9.52 | 82.25 |
| 4 | 14.21 | 14.02 |
| 3 | 18.86 | 2.80 |
| 2 | 17.55 | 0.00 |
| 1 | 39.86 | 0.93 |

may have some issues or complaints in terms of app energy consumption, they also receive positive feedback for certain aspects or features that contribute to their energy efficiency. Briefly, even popular apps can face energy consumption issues that might negatively impact the user experience with a mobile application.

## RQ2: correlation

To investigate whether there is a correlation between the given app ratings and the app reviews related to energy consumption, a quantitative-based analysis have been conducted. Table 5 shows the percentage distribution of given ratings for both positive and negative reviews across 32 different apps. The first column, Rating, represents the possible rating values range from 1 to 5, where one represents the lowest rating, and five represents the

highest rating. The remaining columns, Negative reviews (%) and Positive reviews (%), indicate the percentage of negative and positive reviews corresponding to each rating, respectively. For example, 39.86% of reviews with negative comments received a rating score of 1.

Rating percentages for negative reviews, in which the content expresses dissatisfaction with app energy consumption, as presented in Table 1, show that 57.41% of the users give a rating score of 1 or 2. This suggests that the primary focus of these reviews is to voice complaints specifically about the app's energy consumption. Conversely, the remaining 42.59% of user reviews have been assigned a rating score of 3 or more. This indicates that, despite concerns regarding energy consumption, they may have considered other aspects of the app, such as functionality, performance, and distinct features, to be more noteworthy.

Rating percentages for positive reviews presented in Table 5 imply a connection between user contentment with the app's energy consumption and their propensity to assign a higher rating score, notably a rating of 5, as 82.25% of user reviews have been assigned a rating score of 5. This correlation suggests that users who view the app's energy usage positively are more inclined to provide favorable feedback by assigning the highest ratings. Furthermore, it is evident that the emphasis is primarily on energy consumption, rather than on the other features of the application. This is also supported by the near absence of low or medium scores, such as 1, 2, or 3, being given. Another point to note is that sometimes, the given rating score and the content of the review may not match. A user might accidentally give a rating of 1 instead of 5 or misunderstand that the highest score is 1. Therefore, the fact that 0.93% of the positive user reviews are associated with a rating of 1 could be attributed to such instances of rating inconsistency. For example, the following user review, *"Nice app it doesn't uses many battery and data that's why I like it"* received a rating score of 1.

When assessing the collective energy-related app ratings, none of the apps received ratings of 4 or higher in the overall context of combined energy-related app ratings, as illustrated in Table 6. For the purpose of recall, ratings of 4 or higher were one of the selection criteria for these popular apps. The CapCut, Opera Browser, and Wattpad apps received the highest rating scores of 3.36, 3.24, and 3.14, respectively. Conversely, the Messenger, Google Chrome, and Uber apps obtained the lowest rating scores, registering 1.74, 1.93, and 1.93, respectively. Upon analyzing the energy-related app ratings for all of the considered applications, it is found that the combined average energy-related app rating stands at 2.52 out of 5. This rating score is significantly low compared to the average rating of the considered popular apps, which is 4.37, suggesting the need for enhancements in addressing concerns related to app energy consumption.

Recognizing that users value not only functionality but also responsiveness and resource optimization in an app, the implementation of energy-efficient practices holds the potential to reinforce positive user perceptions. Even in the case of already well-rated apps, incorporating energy-efficient strategies can result in incremental improvements in user review ratings. Users may respond positively to applications that not only meet their

**Table 6 Ratings based on app energy related reviews.**

| Application name | Ratings | Application name | Ratings |
|---|---|---|---|
| Adobe Acrobat Reader | 2.77 | Netflix | 2.71 |
| AliExpress | 2.32 | Opera Browser | 3.24 |
| Amazon Prime Video | 2.92 | Piscart AI Photo Editor | 2.69 |
| Avast Antivirus & Security | 3.08 | Pinterest | 2.92 |
| Canva | 2.68 | Snapchat | 2.34 |
| CapCut | 3.36 | Spotify | 2.76 |
| Duolingo | 2.93 | Telegram | 2.44 |
| eBay | 2.50 | TikTok | 2.41 |
| Firefox | 2.68 | Twitch | 2.58 |
| Google Chrome | 1.93 | Twitter | 2.30 |
| Google Maps | 2.03 | Uber | 1.93 |
| Google Photos | 1.99 | Wattpad | 3.14 |
| Instagram | 2.20 | Waze Navigation & Live Traffic | 2.84 |
| Messenger | 1.74 | WhatsApp Messenger | 1.95 |
| Microsoft Teams | 2.50 | YouTube | 2.12 |
| Microsoft Word | 2.88 | Zoom | 2.24 |

functional needs but also demonstrate a commitment to efficiency and resource optimization.

In practice, this incremental improvement could manifest as users upgrade their reviews from, for example, ratings of 1 and 2 to the highest scores, such as 4 and 5. Notably, the majority of app energy-related reviews are negative, as shown in Fig. 1, and 96.27% of positive app energy-related reviews receive ratings of 4 or 5, as shown in Table 5. Such positive feedback not only contributes to an enhanced app reputation but also fosters a more satisfied and loyal user base, ultimately benefiting the app's overall success and longevity in the competitive app market. In the case of popular apps, with an average rating of more than 4 and the expectation of encountering fewer issues, the impact of energy efficiency enhancements on their already high ratings may be less pronounced. Nevertheless, significantly low energy efficiency-related app ratings may guide developers in their efforts to enhance energy efficiency and, consequently, their app ratings.

In addition to that, while a correlation can be observed between positive reviews and their corresponding score ratings, a precise correlation cannot be observed between negative reviews and the assigned score ratings.

## RQ3: category and functionality

App store categories help users easily find and browse apps that align with their interests and needs; they also group competitor apps within the same category, enabling users to compare and choose from similar apps more conveniently. The categorization of apps generally reflects their distinctive functionalities and features. Exploring the relationship between user reviews, particularly those with high negative ratios regarding energy

consumption, and expectations based on app categories and functionality may reveal valuable insights. Among the apps listed in Table 4, Waze Navigation & Live Traffic, Avast Antivirus & Security, Zoom, Firefox, and Microsoft Teams apps have the highest number of negative review ratios compared to the other apps. These apps are in the categories of Map & Navigation, Business, Communication, Business, and Tool, respectively.

Apps in the Map & Navigation category are often associated with high energy consumption due to their usage of GPS functionality. Since there is an app that has lower negative review ratios in the Map & Navigation category, it may not be solely possible to attribute negative reviews to the nature of this category. Furthermore, while the Uber app is categorized in the Map & Navigation category, the Google Maps app is categorized under Travel and Local category as provided in Table 2. It is indeed anticipated that Waze and Google Maps apps would be in the same category, given their more similar functionality, compared to Uber app. Even if Waze and Google Maps apps were in the same category due to their similar functionality, they do not align with expectations in a similar manner; one exhibits high negative ratios in user reviews regarding energy consumption, while the other does not. Additionally, considering that Firefox, Google Chrome, and Opera Browser share the same category and functionality, one might expect similar energy consumption patterns. However, Firefox differs from them in terms of high negative energy consumption-related user review ratios. In the Tool category, Avast Antivirus & Security is the only app, and given the absence of clear expectations for this category in terms of energy consumption, it may not be thoroughly evaluated. For the Business category, both the Zoom and Microsoft Teams apps exhibit a high number of negative review ratios. The high negative review ratios for both the Zoom and Microsoft Teams apps in the Business category are likely attributed to the functionality of the apps as expected, particularly features such as meetings, video conferencing, and online classes, rather than being linked to the app category. Similarly, a high negative ratio might be expected for the YouTube app in the Video Players & Editors category, along with other video platform-based apps in the Entertainment category, owing to their characteristic focus on video content. However, this expectation does not align with the user reviews.

Consequently, the alignment between user reviews with high negative ratios regarding energy consumption and expectations based on app categories and functionality is not consistently accurate.

## RQ4: trend

To address whether there is a changing trend in app energy consumption issues according to the collected reviews, an approach is adopted that involves analyzing the percentage of user reviews with negative feedback regarding energy consumption issues, considering equally spaced time intervals. This approach allows for a systematic evaluation of how app energy consumption issues have been vary over time. The time intervals consist of 15 data points each, ensuring a minimum of 1 week between intervals for all the apps under consideration, given the variation in starting points of app reviews. As an example, the Zoom app's review start date is November-22, while the AliExpress app's review start date is January-18, as indicated in Table 4, for the collected data.

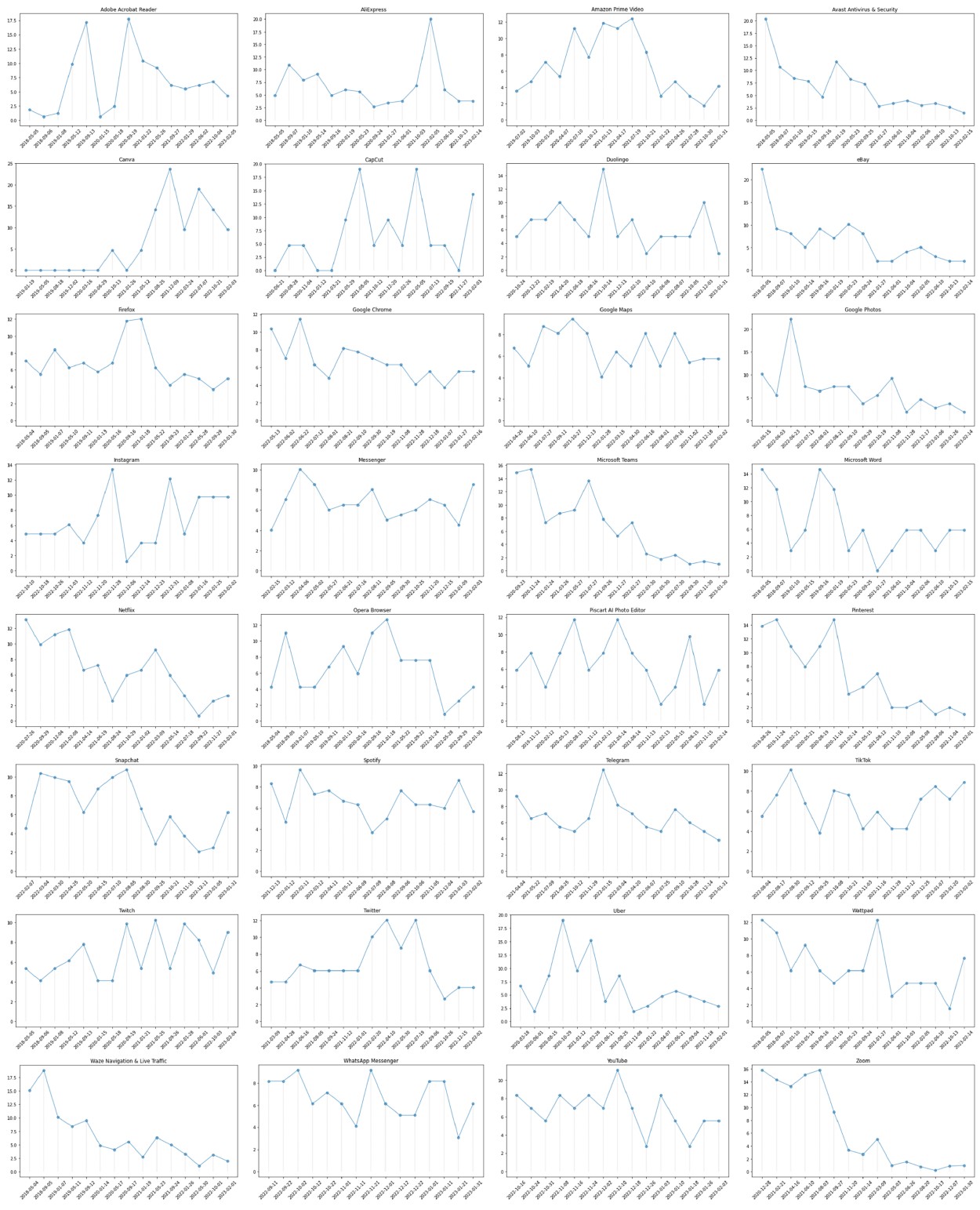

**Figure 2  Trend in energy consumption issues of apps.**

Figure 2 shows the presentation of a plotted line graph for each app, with app names arranged sequentially from left to right. In each plot, the x-axis represents the time intervals, while the y-axis represents the percentage of user reviews with negative feedback for the app. The y-axis values in the figure vary for each individual app represented in the visualization. Based on the data, it is possible to categorize the trends in energy consumption issues of apps as either decreasing or fluctuating. For most of the apps, specifically 25 out of 32, there is a fluctuating trend with variations, but there isn't a clear overall direction. On the other hand, Avast Antivirus & Security, eBay, Google Photos, Microsoft Teams, Pinterest, Waze Navigation & Live Traffic, and Zoom apps exhibit a decreasing trend in energy consumption issues. This may indicate that their energy efficiency may have improved or their usage may have become more optimized. The significance of the result might be further highlighted by the fact that among all the analyzed applications, Avast Antivirus & Security, Microsoft Teams, Waze Navigation & Live Traffic, and Zoom apps have received the highest number of negative reviews, all of which are part of the top 5.

## THREATS TO VALIDITY

One of the most significant threats to the validity is that I considered 32 popular apps from 16 app categories in the Android Play Store. Although the selected apps provide valuable insights, they may not be fully representative of all available popular apps and app categories. The ratings of an app also can vary across different app stores. Therefore, it is possible that an app considered in my study may not meet the criteria to be identified as popular within a different app store. In addition to that, the definition of app popularity can vary depending on the context and perspective.

A more significant threat to validity is the possibility of mistagging. Despite my rigorous efforts to categorize energy related app reviews as positive and negative, there is still a concern about the potential for human error in the tagging process. However, if this concern materializes, the potential impact on the results of the study is expected to be very minimal.

The dynamic nature of app ratings poses another threat. App ratings can change over time, and an app that meets the criteria of being rated equal to or over 4, which is associated with being a popular app, at the time of my study may not necessarily maintain the same rating in the future. However, the findings of this study, based on the collected data at a specific point in time, provide information specifically related to the research question being investigated.

A more specific concern is the possibility that energy-related issues exist within the app, but they are not reflected in user reviews. This concern relates to the presence of unreported or undetected energy issues within the app. However, I believe that this is unlikely because a large sample size of reviews collected generally provides a more comprehensive understanding of users' feedback, thereby increasing the chances of detecting and identifying energy-related issues within the app.

Finally, this study leverages keyword analysis to detect users reviews regarding app energy consumption. Misspellings in user reviews for critical keywords such as "battery",

"energy", and "power" may lead to omission some relevant information during the analysis process. However, the size of the collected data is expected to be at a level where the situation of misspelling in critical keywords can be ignored, and it is anticipated that it will not have a significant impact on the results of the study in such cases.

## RELATED WORK

Software energy efficiency, especially in battery-powered devices, is of utmost significance. As a result, researchers have focused on offering energy-related insights to developers by empirically investigating the underlying causes of energy consumption. These studies include impacts of: performance tips (*Sahin, Pollock & Clause, 2016*), code obfuscation (*Sahin et al., 2016*), code refactorings (*Sahin, Pollock & Clause, 2014*), design patterns (*Sahin et al., 2012*), web servers (*Manotas et al., 2013*), sorting algorithms (*Bunse et al., 2009*), advertisements (*Gui et al., 2015*), API usage (*Linares-Vásquez et al., 2014*; *Manotas, Pollock & Clause, 2014*), and programming models (*Cohen et al., 2012*). Furthermore, researchers have delved into various aspects of an application's energy consumption. This includes exploring energy usage across different versions (*Hindle, 2012*) and distinct implementations of the same application (*Arunagiri et al., 2011*). They have also examined the energy implications of selecting between different software systems with similar purposes (*Amsel et al., 2011*), studied how developers inquire about energy usage (*Pinto, Castor & Liu, 2014*), and investigated developers' perspectives on energy considerations during software development (*Manotas et al., 2016*).

Due to the limitations of battery-powered devices, the energy efficiency of mobile apps has evolved into a significant criterion in the process of selecting applications with similar purposes. To aid users in their selection process, *Behrouz et al. (2015)* introduced an approach aimed at providing insights into the energy consumption of applications within the same category. Remarkably, following a methodology similar to that employed in this study, *Saborido et al. (2016)* also conducted a comparative analysis of applications within the same category, with a focus on their energy consumption.

In more recent studies, *Palomba et al. (2019)* have explored the influence of Android-specific code smells on energy consumption, finding that refactoring these code smells consistently leads to a reduction in energy consumption across various scenarios. An Android Studio plugin, EcoAndroid, has been introduced with the purpose of streamlining the development of energy-efficient mobile applications by automatically implementing distinct energy-related refactorings (*Ribeiro, Ferreira & Mendes, 2021*). Similarly, the GreenSource infrastructure is designed to characterize energy consumption in the Android ecosystem, offering both Android developers and researchers a platform for contemplating energy-efficient Android software development (*Rua, Couto & Saraiva, 2019*). To optimize the energy consumption of mobile games, *Choi et al. (2022)* introduce a system-level energy optimization scheme for game applications on EAS-enabled mobile devices, incorporating features such as an Lsync-aware GPU DVFS governor, adaptive capacity clamping, and on-demand touch boosting. Byte-code transformations combined with genetic search can also be used to reduce an app's energy consumption (*Bangash, Ali & Hindle, 2022*). To assist in the development process of Android apps, an approach is

proposed to address the oracle problem in testing the energy behavior of mobile apps (*Jabbarvand, Mehralian & Malek, 2020*). The results of the study show that the proposed approach is able to provide accurate and fast detection of the existence of energy defects. *Neves et al. (2023)* investigated the recently introduced single-activity app architecture for Android apps, characterized by a fundamentally different structure and runtime behavior during screen transitions. The study asserts a significant difference in energy consumption between the single-activity and multiple-activity architectures. With the advent of COVID-19, online meeting apps have become more popular. *Wattenbach et al. (2022)* have compared energy consumption of Google Meet and Zoom apps, assessing factors like the number of call participants, microphone and camera use, and virtual backgrounds. Their findings suggest that Zoom app is more energy-efficient than Google Meet app.

Keywords and adjust words concerning energy-related issues in user reviews of mobile apps have also been subject to investigation (*Phong et al., 2015*; *Wilke et al., 2013*). These studies provided a list of words potentially linked to energy-related problems mentioned within reviews. In addition to that, *Wilke et al. (2013)* suggested that The Economist and The Weather Channel are popular apps that have energy consumption issues based on user reviews.

Examining user reviews on Google Play Store to assess different aspects of the app, apart from energy consumption concerns, has also been another focus of research studies. For example, *Aljedaani et al. (2022)* focused on assisting developers by automatically uncovering accessibility challenges that adversely affect the user experience for individuals with disabilities in mobile apps through the analysis of user reviews. The relationship between user reviews and security and privacy-related changes in apps has been explored (*Nguyen et al., 2019*). Additionally, *Wang et al. (2020)* introduced an automated process for identifying functionality-relevant user reviews and inferring their permission implications. *Gao et al. (2019)* focused on identifying emerging issues through user reviews. Moreover, a distinct study addressed automated bug reproduction through user reviews (*Li et al., 2020*). In another study, the objective was to offer users insights into sentiment analysis outcomes of user reviews and to categorize applications into specific groups based on the most favorable sentiment analysis results (*Setiawan & Mawardi, 2022*).

## CONCLUSIONS

In this study, I delved into the critical question of whether popular apps exhibit issues concerning energy consumption. A total of 32 apps were carefully selected, representing 16 distinct app categories within the Google Play Store. These apps were selected from the realm of popular apps, meeting the criteria of possessing ratings of 4.0 or higher, accumulating over 3 million reviews, and achieving more than 100 million downloads. A total of 14,064 user reviews containing the specified keywords from the collected data (3,818,950 user reviews) were manually examined. Among these, 8,007 reviews were identified as directly related to the app's energy consumption and categorized as negative or positive reviews. The result of this study demonstrate the following:

- Even popular apps have energy consumption issues that negatively affect user experience.
- Positive reviews exhibit a correlation with score ratings, but this correlation is not observed in the case of negative reviews concerning app energy consumption.
- Enhancing app energy efficiency has the potential to improve user satisfaction and, consequently, contribute to the overall success of the app.
- Linking energy consumption-related issues only with app categories and even similar app functionalities is unlikely since it is not consistently accurate with expectations.
- With the exception of specific apps, a fluctuating trend with variations characterizes app energy consumption issues.
- Out of the top five apps with the highest number of negative reviews, four exhibit a decreasing trend in energy consumption issues.

Indeed, the energy efficiency of an app stands as a important consideration for users, even in the case of popular apps that offer a wide range of functionalities and features. Therefore, developers need to attentively address user expectations and optimize their apps for energy efficiency.

On the basis of these conclusions, there are several potential areas for future work. First, I plan to extend this study by comparing app reviews for the same app that is considered popular in different app stores. This comparative analysis would provide insights into potential variations in user perceptions and feedback regarding energy consumption issues across different platforms. Second, I plan to develop an automated framework specifically designed to extract and analyze energy-related feedback from app reviews. This framework can assist developers in proactively identifying and resolving energy-related issues, ultimately leading to improved energy efficiency and user satisfaction.

### Funding
The author received no funding for this work.

### Competing Interests
The author declares that they have no competing interests.

### Author Contributions
- Cagri Sahin conceived and designed the experiments, performed the experiments, analyzed the data, performed the computation work, prepared figures and/or tables, authored or reviewed drafts of the article, and approved the final draft.

### Data Availability
The data is available at FigShare: Sahin, Cagri (2023). PlayStoreData. figshare. Dataset. https://doi.org/10.6084/m9.figshare.24645237.v1.

## Supplemental Information

Supplemental information for this article can be found online at http://dx.doi.org/10.7717/peerj-cs.1891#supplemental-information.

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
