# Peer review of "Do popular apps have issues regarding energy efficiency?"

_PeerJ Computer Science, doi:10.7717/peerj-cs.1891_

## Round 0.1 · original submission · Major Revisions

The reviewers both give good comments to what should be improved in the paper. I would summarise them as follows:

- The correlation between positive/negative energy efficiency reviews and rating is too simple. Use a better analysis to quantify the actual impact of these reviews.

- RQ 3 is difficult to answer. When would you answer it with "no"? This RQ should be rethought.

- The manual categorisations need additional validation.

- Please make the raw data available or give a clear statement in the paper why this is not possible.

I am looking forward to your revision!

Reviewer 1 ·

Basic reporting

Overall, the ideas in this paper are very interesting and practical. The English writing of the paper is very good, easy to read and easy to understand. However, the methodology of this paper appears to be simple, only keyword search analysis was performed on review data. Therefore, it is difficult to assess the innovativeness of this paper.

Experimental design

The number of comments related to energy efficiency is very small compared to the total review data. Some apps have only about one in a thousand reviews related to energy efficiency. Thus, the ratings corresponding to these reviews have a very limited impact on the average score. How to evaluate this impact needs further elaboration and more proof.

Validity of the findings

no comment

Additional comments

There are some minor typos in the text. For example, some of the data in Table 2 are missing units, 14.3, 500+, 100+, etc.

Cite this review as

·

Basic reporting

This study investigates the impact of energy consumption issues on popular mobile apps with high user ratings. Analyzing 14,064 user reviews from 32 widely used apps across 16 categories reveals that all examined apps exhibit energy-related concerns. The results emphasize the importance of addressing energy efficiency in app development to meet user expectations, enhance satisfaction, and potentially improve app ratings.

+ Exceptionally crafted and engagingly presented
+ Addresses a contemporary issue with a compelling idea and methodology
+ Offers insightful perspectives on the evolution of apps concerning their energy efficiency.
+ The software used for collecting app review data is intriguing.
+ The impressive dataset size adds to this work's overall contribution.

- In my opinion, addressing RQ3 proves challenging
- The categorization process could benefit from additional validation
- Certain data visualizations could be enhanced
- Raw data is notably absent.

Literature references, sufficient field background/context provided:

The article does cover many of the leading literature references on the field of green software, in particular on Android energy consumption. However, most references are pretty old (from 2016 or earlier). I'd like to see some more recent references here.

Professional article structure, figures, tables. Raw data shared:

I've some things here that IMHO could be improved:
Figure 1: This figure is quite complex to follow in the current state. It's pretty reasonable to point out that most reviews are negative, but it's hard to figure out the exact percentual number of positive reviews. IMHO, this figure could be improved to make it easier to see the data.
Table 5: third row, third column, there's an additional 'd'
Lines 348-351: "(...) is found that the combined average energy-related app rating stands at 2.52 out of 5." I'd like to see this data in the article. The combined energy-related rating seems like an exciting way to perceive how the averages see the energy efficiency of these apps.
Figure 2: It should be noted that the y-axis on these figures differs.
The data collection software source code was shared, but not the raw data. This seems like a lapse because the folders that should contain the raw data are present in the supplemental files.

Experimental design

Methods described with sufficient detail & information to replicate:

Lines 236-237: ""To gather the data necessary to answer my research questions, the keywords “battery”, “energy”, and “power" were scanned in the collected app reviews for each app."" it's not clear to me if these were the keywords here were used with or without the adjunct words

Lines 292-293: "As a result, I have decided to manually tag the energy consumption-related app reviews as positive and negative." While I concur with the author's preference for manual tagging of app reviews, the potential for disagreement on specific tag assignments raises a concern. Clarity is needed on whether all tags (or even a subset of them) went under subsequent validation by an independent specialist.

RQ3 - "Can app categories be effectively classified based on variations in the frequency of energy consumption-related issues?"

IMHO, this seems like a tricky question to answer because of the number of apps in each category presented in the study. 'Communication' is the category with the most apps (7), followed by 'Entertainment' and 'Social' (3 each), then 'Productivity', 'Shopping', 'Video Players & Editors', 'Photography', 'Business', 'Map & Navigation' (2 each). The rest of the categories only have a single app. The challenge lies in extrapolating findings from a limited app subset to the extensive diversity within each category comprising thousands of apps. I would like to see this point further clarified.

Validity of the findings

All underlying data have been provided; they are robust, statistically sound, & controlled.

As stated before, I was unable to find the data.

Cite this review as

---

## Round 0.2 · accepted · Accept

You addressed all previous comments from the reviewers and my satisfactorily. The manuscript is now ready for publication.

Reviewer 1 ·

Basic reporting

This paper is written in good English and is easy to read and understand.

Experimental design

The experimental process is clearly described and the experimental results are detailed.

Validity of the findings

No comment.

Additional comments

The author has revised all the comments.

Cite this review as

·

Basic reporting

The author have answered all my questions in a satisfactory manner and IMHO is fit for publication.

Experimental design

No comment

Validity of the findings

No comment

Cite this review as